# A Comparative Genomic Analysis of Epstein–Barr Virus Strains with a Focus on EBV2 Variability

**DOI:** 10.3390/ijms26062708

**Published:** 2025-03-17

**Authors:** Ana Catalina Blazquez, María Dolores Fellner, Mario Alejandro Lorenzetti, María Victoria Preciado

**Affiliations:** 1Laboratorio de Biología Molecular, División Patología, Instituto Multidisciplinario de Investigaciones en Patologías Pediátricas (IMIPP), CONICET-GCBA, Buenos Aires C1425EFD, Argentina; blazquez.a.catalina@gmail.com; 2Servicio Virus Oncogénicos, Laboratorio Nacional de Referencia de Virus Epstein-Barr, Departamento Virología, Instituto Nacional de Enfermedades Infecciosas (INEI)—ANLIS, “Dr C. Malbrán”, Buenos Aires C1425EFD, Argentina; fellnermd@gmail.com; 3Consejo Nacional de Investigaciones Científicas y Técnicas (CONICET), Buenos Aires C1425FQB, Argentina

**Keywords:** Epstein–Barr virus, EBV2, sequence variation, South America

## Abstract

Most genomic studies on Epstein–Barr virus variability have focused on the geographic and pathological associations of EBV1 genomes. In contrast, the variability of EBV2 genomes has been less explored, mainly due to their restricted geographic circulation and the lesser number of sequenced EBV2 isolates. In this study, we sequenced and analyzed twenty-eight EBV1 and ten EBV2 genomes and a potential recombinant from Argentina, which were combined with two-hundred-and-thirty-nine downloaded complete genomes from other geographic regions, to produce an initial *multi-sample.vcf* file comprising 278 EBV genomes. In this context, we identified 1093/4541 positions in the viral genome that contribute to variability between viral types, mainly located in the EBNA2 and EBNA3 family of genes and the adjacent BZLF1, BZLF2, and BLLF1 genes. We further described that this variability exhibits distinct patterns across Africa, South America, and Southeast Asia. Compared to EBV1 genomes, EBV2 genomes showed fewer variable positions relative to their reference genome (Wilcoxon test, *p* = 0.0001). Principal component analysis revealed that EBV2 genomes from Southeast Asia segregate independently from those from South America (Wilcoxon test, Bonferroni correction; *p* = 1.1 × 10^−7^) and Africa (Wilcoxon test, Bonferroni correction; *p* = 2.6 × 10^−9^). Additionally, we identified those precise variable positions with geographic segregation strength: 1135/3666 in EBV1 and 380/3276 in EBV2. Furthermore, the distribution of variable positions along the genome disclosed a close relation for EBV2 isolates from Africa and South America as compared to isolates from Southeast Asia. Although our analysis is limited to EBV2 genomes isolated from three geographic regions, this was, to the best of our knowledge, the first study to comprehensively characterize the geographic variability of the complete EBV2 genome. These findings underscore the geographic and genetic diversity of EBV2 genomes and contribute to understanding the EBV’s evolutionary dynamics and potential regional adaptations. This research enhances our understanding of EBV2 genomic variability, supporting future epidemiological studies and advancing the knowledge base for targeted treatments and vaccine development for EBV-associated diseases.

## 1. Introduction

The Epstein–Barr virus (EBV), taxonomically classified as *Lymphocryptovirus humangamma4*, is one of the most widely distributed human viruses, with a prevalence exceeding 95% in the adult population [1].The EBV is the etiological agent of infectious mononucleosis; which may develop after primary infection and presents with a higher incidence in developed countries in comparison to developing ones, where primary infection occurs in childhood and is usually asymptomatic [2,3]. While acting as a harmless passenger in most cases, it may also behave as a tumor driver co-factor [4]. In this sense, the EBV is associated with lymphoid neoplastic processes such as Hodgkin’s lymphoma (HL), Burkitt’s lymphoma (BL), diffuse large B-cell lymphoma (DLBCL), and post-transplant lymphoproliferative disorder (PTLD), as well as with epithelial tumors like in gastric carcinoma (GC) and nasopharyngeal carcinoma (NPC) [5,6].

Based on sequence variations mainly in EBNA2 and the EBNA3 gene family, the EBV is classified into two types, EBV1 and EBV2; however, genomic variation exceeds this set of genes [1]. Moreover, each EBV type displays its own geographical circulation pattern; while EBV1 is globally distributed and is generally more prevalent than EBV2 across most regions, in Sub-Saharan Africa, the prevalences of both viral types are similar. Conversely, certain tropical areas such as Indonesia, Papua New Guinea, and Northern Brazil exhibit a higher circulation of EBV2 genomes. Although the grounds for the higher prevalence of EBV2 in tropical regions are not fully understood, a potential selective advantage for this viral type in populations chronically exposed to immunomodulating infections, like endemic malaria or other tropical diseases, may exist [7,8]. Beyond its distinct geographic distribution, EBV2 has been identified more frequently in immunosuppressed individuals in any geographic region, such as in patients infected with the human immunodeficiency virus (HIV) or individuals with inborn errors of immunity [9,10]. Regarding virus–host interactions, EBV1 has a greater capacity to transform B-lymphocytes in culture compared to EBV2 due to amino acid differences in the transactivation domain of EBNA2 [11]. Another phenotypic difference observed between both viral types is that EBV2 in vitro infected B cells exhibit a higher expression of lytic proteins compared to those infected with EBV1 [12,13]. This latter difference could be attributed to the presence of the Zp-V3 variant of the BZLF1 gene promoter in all EBV2 genomes whereas the prototypical Zp-P (also referred as Zp-V1) variant predominates in EBV1 [7]. The Zp-V3 variant in the promoter region of the BZLF1 gene contains an additional binding site for the cellular transcription factor NF-AT, which increases the transcription of the viral gene, whose protein product Zta plays a crucial role in inducing the switch from viral latency to the lytic cycle [14].

Recent advances in the number of complete EBV genomes and the development of bioinformatics tools have enabled the more precise classification of viral variants and types. Genome-wide single nucleotide polymorphism (SNP) comparison between viral types has demonstrated that although sequence differences are mainly located in the EBNA2 and the EBNA3 gene family, variation also extends to the BLLF1 and BZLF2 genes, which encode for the membrane glycoproteins gp350/220 and gp42, respectively [15,16]. Even though viral types represent the largest natural variation within the EBV genome, several authors have observed differential geographical circulation of genomic variants within EBV1 [17,18]. Moreover, in a previous report, our group further identified the variable positions that contribute to this geographic segregation in EBV1 genomes [19]. However, the variability in EBV2 genomes has been less explored due to the smaller number of sequenced genomes, a fact that could be attributed to their restricted geographical distribution. Given that the proportion of EBV2 genomes circulating in South America is steadily increasing [8,20], this study aimed to extensively investigate the variability of EBV2 genomes in a geographical context and compare it with EBV1 variability.

## 2. Results

Complete EBV genome sequencing was performed in isolates from Argentina and analyzed together with raw data from different geographic origins to produce a viral-type- and geography-balanced dataset. Viral typing and recombination analysis involving EBV types was assessed and two-hundred-and-six genomes were typed as EBV1, sixty-seven as EBV2, and five as potential intertype recombinants; in particular, twenty-eight Argentine genomes were classified as EBV1, ten as EBV2, and one as a potential recombinant (see Appendix A). Intertype recombinations were confirmed through the de novo assembly and further phylogenetic reconstruction of the EBNA2 gene and the EBV1 and EBV2 references showed that four isolates segregated with the EBV1 reference and one segregated with the EBV2 reference (see Appendix A). Additionally, the same strategy was also implemented for the EBNA3 family of genes, revealing that four isolates clustered with the EBV2 reference while the other one segregated with the EBV1 reference (see Appendix A). Based on these findings, four genomes represented EBV1–EBV2 recombinants (ERS549494, ERS549475, ERS17911187, and ERS1791207) while the remaining one, an Argentinean isolate, represented an EBV2–EBV1 recombinant (SRR29321001). Finally, unequivocal recombination events were confirmed and precise braking-point locations were assessed with RDP4 (see Appendix A).

### 2.1. Variability Between EBV1 and EBV2 Genomes

As expected, based on Principal Component 1 (PC1 = 20.23%) in the Principal Component Analysis (PCA), EBV1 genomes were significantly segregated from EBV2 genomes (Wilcoxon test, *p* value < 2.2 × 10^−16^) (Figure 1a). The intertype variability of the complete EBV genomes was further analyzed by PCA using a *multi-sample_all genomes*.vcf file, which contained 4541 variable positions (4412 SNPs and 129 INDELs), as an input file (See Figure 6 in Section 4.6).

Given that the strong linkage disequilibrium between variants in the EBNA2 gene and the EBNA3 gene family with the EBV type may not be the only type-specific variations along the genome, we sought to explore the presence of other characteristic polymorphisms contributing to the segregation of viral types. In this second analysis, the contribution of the PC1 of each of the 4541 variable positions between EBV1 and EBV2 genomes was evaluated using Fisher’s exact test to quantify the contribution of each variable position to the segregation of viral types (Figure 1b,c). As observed in Figure 1b, the further a value is from the ordinate axis, the greater the contribution of that variant is to PC1, which is reflected in increasingly significant *p* values. In this way, 1093 variable positions showed statistical strength to segregate both viral types. Figure 1c represents the distribution of these variable positions along the viral genome, as well as the significance of their contributions to the segregation of viral types, based on the −log_10_(*p* value) obtained. In this way, from a total of 58 variable positions in EBNA2, 21 (36.20%) showed statistical strength to segregate viral types. Similarly, 542 from 819 (66.17%) were characterized in the EBNA3 family of genes and 24/59 (40.67%) in BZLF1, 12/19 (63.15%) in BZLF2, and 38/79 (48.10%) in BLLF1 genes (see Appendix A). Although the major genetic differences among viral types lie within the EBNA2 gene and the EBNA3 gene family, intertype variability also extends into the region adjacent to the EBNA3 genes, particularly affecting the BZLF1, BZLF2, and BLLF1 genes.

Regarding the BZLF2 gene, which codes for glycoprotein gp42, significant differences were observed between both viral types at 12 variable positions within the coding region. Among these, five missense variants were exclusively harbored by EBV2 genomes: Ala38Ser, Val76Ile, Gln92Lys, Gly113Glu, and Cys114Arg (see Appendix A). Similarly, for the BLLF1 gene, coding for glycoprotein gp350/220, 36 variable positions with significant differences between both viral types were identified. Polymorphisms in the N-terminal region of this protein exhibited the most significant type-specific segregation strength, where Pro31Leu, Val48Ala, Gln60Lys, Asp62Asn, and Gln67Leu were exclusively found in EBV2 genomes while His17Gln, Gly20Arg, Glu21Asp, Thr38Ala, and Val115Ile were more prevalent in EBV2 but also found in EBV1 from OCE (see Appendix A). With respect to the promoter region of the BZLF1 gene, four EBV2 genomes failed to fully cover this region; in consequence, the Zp-V3 variant (91,006 A > C, 91,012 T > C, and 91,047 T > C) was found in 64/206 EBV1 genomes and in 63/63 EBV2 isolates. On the other hand, the Zp-P variant was identified in 142/206 EBV1 isolates and was absent in EBV2 isolates, thus implying a strict association of the Zp-V3 variant with EBV2 genomes (Fisher’s exact test; *p* value < 2.2 × 10^−16^). Finally, on the coding region of the BZLF1 gene, coding for Zta protein, nine missense variable positions and one disruptive in-frame deletion with significant segregating strength between both viral types were found: Thr68Ala, Ser76Pro, Ala127del, Ala135Thr, Val146Ala, Val152Ala, Gln163Leu, Glu176Asp, Gln195His, and Ala205Ser. While Thr68Ala, Ser76Pro, Ala135Thr, Val146Ala, Val152Ala, Gln163Leu, Glu176Asp, Gln195His, and Ala205Ser were more prevalent in EBV2 genomes, Ala127del was a hallmark deletion of EBV1 (see Appendix A). In summary, BZLF1, BZLF2, and BLLF1 genes all displayed EBV2-type-specific variable positions.

In order to assess the variability between EBV1 and EBV2 genomes isolated in each geographic region, three additional *multi-sample_all genomes.vcf* files based on the geographical origins of the isolates (*multi-sample_SAM.vcf, multi-sample_SEA.vcf* and *multi-sample_AFR.vcf*) were constructed and used as input for PCA. Within each region, intertype variability was evaluated and 870/3048 variable positions that differentiated between both viral types in AFR (see Appendix A), 509/2829 in SAM (see Appendix A), and 531/2679 in SEA (see Appendix A) were characterized. As described above for the entire dataset, intertype segregation potential remained in the EBNA2 gene, the EBNA3 gene family, and the adjacent BZLF1, BLLF1, and BZLF2 genes when assessing geographic regions independently. Even though type segregation potential was retained by the latter genes across all the geographies, variable positions in the BZLF2 gene (Ala38Ser, Val76Ile, Gln92Lys, Gly113Glu, and Cys114Arg) were not capable of discriminating sequences by geographic origin. On the contrary, amino acid changes in the C-terminal domain of the gp350/220 protein (Asp441Ala, Trp495Arg, Asn672Ile, Ile710Thr, Gln754Lys, and Pro812Thr) showed a higher prevalence among EBV2 isolates from Africa. On the other hand, BZLF1 displayed the greatest type segregation potential in each region. Regarding the gene’s coding region, the missense variable position resulting in Glu176Asp change showed significant differences between EBV1 and EBV2 in the three geographic regions (0/33 in EBV1 and 10/16 in EBV2 in SEA; 0/36 in EBV1 and 37/38 in EBV2 in AFR and 0/31 in EBV1 and 10/10 in EBV2 in SAM), meaning an association of EBV2 isolates with Glu176Asp (Fisher’s exact test; *p* value < 2.2 × 10^−16^). Since Glu176Asp is a hallmark of the BZLF1-A2 haplotype, this haplotype resulted in association with EBV2 across all geographies. Unlike in SEA, where only the Glu176Asp amino acid showed segregation strength for viral types, in AFR and SAM, the amino acid changes in Thr68Ala, Ser76Pro, Val152Ala, Gln163Leu, Glu176Asp, Gln195His, and Ala205Ser also displayed type-specific segregation strength and occurred preferentially in EBV2 genomes.

From the *multi-sample.vcf* files, the Zta protein haplotypes of each viral genome were identified. This analysis showed that 100% of EBV2 genomes (67 isolates) contained the BZLF1-A haplotype. Conversely, among EBV1 isolates, the BZLF1-A (17/33) and BZLF1-B (13/33) haplotypes were more prevalent in SEA while the BZLF1-C haplotype was the most common in AFR (20/36) and the BZLF1-B haplotype was the most common in SAM (17/31). With respect to the BZLF1 gene promoter region, all EBV2 genomes harbored the Zp-V3 haplotype; however, the association of the Zp-V3 haplotype with EBV1 differed among the studied geographic origins. This haplotype showed a higher prevalence among SEA EBV1 (17/33) isolates with respect to AFR (7/36) and SAM (5/30) genomes (Fisher’s exact test; *p* value = 0.00096).

Additional discrepancies between EBV1 and EBV2 isolates were also observed across the rest of the viral genome, and these differences were grouped into three distinct blocks other than the above-mentioned genes, as indicated in Figure 2. The first block extended from 46,603 to 74,593 bp and included the coding region of genes BFRF1, BFRF2, BFRF3, BPLF1, BOLF1, BORF1, BORF2, BaRF1, BMRF1, BMRF2, BSLF2/BMLF1, and BSLF1. The variability within this block varied across the three geographic regions, with a higher number of variable positions (109/473) observed in the AFR isolates and a lesser number (1/392) in SEA ones (Fisher’s exact test; *p* value: <2.2 × 10^−16^). On the other hand, SAM genomes harbored 19/432 variable positions that significantly segregated EBV2 from EBV1. While variable positions contributing to intertype differentiation were distributed throughout the entire block in AFR genomes, in SAM genomes, variability was preferentially concentrated around 50,000 bp and was predominantly located within the coding regions of the BFRF2, BFRF3, and BPLF1 genes (Figure 2).

The second block comprised nucleotides from 104,403 to 112,651 bp and displayed a lower number of variable positions that contributed to intertype differentiation across the three geographic regions; however, variable positions with type-segregating capacity were made evident in AFR (6/143), in SEA (9/121), and in SAM (2/113). Specifically, in SAM and AFR, variability was primarily concentrated in the BBLF2/BBLF3 gene. In contrast, in SEA, variable positions contributing to type diversity were mainly located within the BGLF4 and BGLF3 genes (Figure 2).

Finally, the third block encompassed variable positions (60/397 in AFR, 20/296 in SEA, and 0/324 in SAM) with type segregating strength values between 151,703 and 169,016 bp (Fisher’s exact test; AFR vs. SEA *p* = 0.0007); conversely, no significant differences were found between the two viral types in SAM isolates within this region. Among AFR and SEA isolates, these type segregating variable positions were mainly located within BARF0 and LMP1, but while AFR genomes contained variable positions distributed throughout the entire block, SEA isolates concentrated their type segregating variability near the terminal repeat (Figure 2).

### 2.2. Comparative Geographic Variability Analysis in Each EBV Type

To further explore intratype variability in the geographic context, a new *multi-sample_EBV1.vcf* file comprising 206 EBV1 genomes with 3666 variable positions was constructed and used as an input file for PCA in order to assess the geographic segregation strength of variable positions in the EBV1 genomes. The relationship of each EBV1 genome with the first two principal components explained the segregation of sequences based on the geographical origin of the isolates (Kruskal–Wallis test, *p* value PC1 < 2.2 × 10^−16^, *p* value PC2 < 2.2 × 10^−16^) (Figure 3a). In this way, PC1 (21.62%) explained the segregation of isolates from Oceania (OCE), South Asia (SAS), and Southeast Asia (SEA) from the rest of the world [Africa (AFR), Europe (EUR), Australia (AUS), North America (NAM), and South America (SAM)] while PC2 (10.28%) differentiated isolates from OCE and SEA. Additionally, PC3 (6.35%) also contributed to explaining the segregation of EBV1 genomes based on their geographical origins: in particular, that of isolates from SAM from those from EUR, AUS, and SAS (Kruskal–Wallis test, *p* = 1.06 × 10^−11^) (see Appendix A).

In a similar way, to study the variability of EBV2 genomes in a geographical context, a *multi-sample_EBV2.vcf* file was constructed from 67 EBV2 genomes and the EBV2 reference genome including 3276 variable positions (3155 SNPs and 121 INDELs) and was used as the input file for PCA. The geographical origins of EBV2 genomes were mainly restricted to AFR (38), SAM (10), and SEA (16) while only one each was present in OCE, EUR, and AUS and none in SAS and NAM. In this PCA analysis, PC1 was able to discriminate EBV2 genomes according to geographical origin (Kruskal–Wallis test; *p* = 1.37 × 10^−9^) (Figure 3b). In this way, PC1 (25.88%) allowed the differential segregation of genomes from SEA compared to genomes from AFR (Wilcoxon test, Bonferroni correction; *p* = 2.6 × 10^−9^) and SAM (Wilcoxon test, Bonferroni correction; *p* = 1.1 × 10^−7^); however, it did not explain the variability between AFR and SAM genomes (Wilcoxon test, Bonferroni correction; *p* = 1). Additionally, PC1 revealed two subpopulations of EBV2 in Africa since 18 genomes presented with PC1 values close to 0 [median: 0.93; (Min.: −1.13–Max.: 3.44)] and 19 genomes had PC1 values > 0 [median: 9.48; (Min.: 8.02–Max.: 9.74)], a phenomenon not observed in the PCA derived from EBV1 genomes. On the other hand, PC2 (14.83%) contributed less in explaining the geographical segregation of EBV2 genomes (Kruskal–Wallis test; *p* = 0.0027); similarly to PC1, it allowed the segregation of genomes from SEA with respect to isolates from AFR and SAM (Wilcoxon test, Bonferroni correction; *p* = 0.023), but could not explain the variability between AFR and SAM genomes (Wilcoxon test, Bonferroni correction; *p* = 1) (Figure 3b).

To further explore the observed geographic variability, the number of variable positions relative to the reference genomes of EBV1 and EBV2 were quantified using the information from each *multi-sample.vcf* file. A significant difference was observed between both viral types (Wilcoxon test, *p* = 0.0001), with EBV2 showing less variability relative to its reference genome as compared to the variability of EBV1 isolates relative to the EBV1 reference. When considering each geographical origin independently, no significant differences in the median number of variants between viral types were found in SAM (Wilcoxon test, *p* = 0.12); however, in AFR and SEA, the median number of variants in EBV2, relative to its reference, was lower than that observed for EBV1 (Wilcoxon test, *p* AFR = 0.019 and *p* SEA = 0.0029) (Figure 4a). In addition, the number of variable positions relative to the corresponding reference genome showed significant differences among geographic regions for both viral types (Kruskal–Wallis test; *p* EBV1 = 3.69 × 10^−10^; *p* EBV2 = 0.00010). In this context, EBV genomes from SEA, regardless of the viral type, showed a higher number of variable positions in comparison to that observed in isolates from AFR (Wilcoxon test; *p* EBV1 = 3.05 × 10^−9^; *p* EBV2 = 0.00082) and SAM (Wilcoxon test; *p* EBV1 = 1.61 × 10^−7^; *p* EBV2 = 0.00010) (Figure 4a). Common variable positions, present in more than 75% of the sequences, were plotted in relation to their positioning in the genome to further analyze and compare their precise location along each type-specific genome in each geographic region (see Appendix A, Figure 4b,c). With these settings, variable positions in SEA EBV1 genomes were distributed evenly throughout the entire viral genome; however, in SEA, EBV2 genomes variable positions were restricted to discrete regions in the genome (Figure 4c). On the other hand, variability patterns observed in AFR and SAM were similar although type-specific. While, in EBV1 genomes, both AFR and SAM exhibited significant variability between 75,000 and 110,000 bp, EBV2 genomes concentrated the majority of variable positions in the region adjacent to 50,000 bp and the region between 110,000 and 150,000 bp.

To unequivocally identify those precise variable positions that contributed to the segregation of the genomes in relation to their geographic origins, the distribution and significance of each variable position in the *multi-sample_EBV1.vcf* and *multi-sample_EBV2.vcf* files were assessed for both viral types by a PCA (Figure 5).

As described above in Figure 3a and based on PCA results that showed that PC1 segregated two main geographical groups (SAS–SEA–OCE vs. AFR–SAM–NAM–EUR–AUS) and PC2 was able to separate OCE from SEA–SAS, the genomic positioning of these variable positions was further assessed. From the 3666 variable positions considered, 1135 (30.9%) showed the ability to discriminate isolates from SAS–SEA–OCE from the cosmopolitan group (see Appendix A, Figure 5a). When plotting the distribution of variable positions along the viral genome, it was observed that the variability contributing to the diversity between the two geographical groups was distributed throughout the entire viral genome; however, the most significant variants were located in four main regions, namely in the BPLF1, EBNA1, BcRF1 and miR-BART, and LMP1 genes (Figure 5b).

Similarly, precise variable positions contributing to the separation of EBV2 genomes into the previously identified geographic groups (SEA–OCE vs. AFR–SAM) were assessed (Figure 3b). Of the total 3276 variable positions considered for EBV2, 380 (11.5%) showed the capability to discriminate SEA–OCE from AFR–SAM genomes; this number of significant variable positions was smaller than that observed in EBV1 isolates (Figure 5c, see Appendix A). The most capable geography-segregating variants for EBV2 genomes were discretely distributed in three genomic regions, namely those with the BPLF1, EBNA1, and BcRF1 genes (Figure 5d). In particular, variations in the EBNA1, BcRF1, and BPLF1 genes contributed significantly to geographic diversity in both viral types. However, unlike in EBV1, in EBV2, the EBNA3 gene family and adjacent genes did not exhibit differential variability between SEA and OCE as compared to AFR and SEA.

## 3. Discussion

In recent years, with the advent of high-throughput sequencing technologies, more than 1000 complete genomes of the EBV have been sequenced. Since most of the available EBV genomes in public databases correspond to EBV1, most evolution, geographic variability, and pathology association studies have focused on this viral type. Consequently, and due to the smaller number of sequenced EBV2 genomes, its variability has not been fully explored [15,19,21,22]. Since EBV2 appears to be increasingly circulating in Argentina, this study aimed to identify variable genomic regions responsible for the segregation of EBV types and their geographic distribution, focusing on EBV2 variability in a previously underrepresented geographic region. Specifically, previous studies that evaluated the prevalence of both viral types in healthy carriers or children with benign infections demonstrated a circulation that ranged between 85% and 95% in favor of EBV1 [23,24]. On the other hand, only certain tropical regions in South America, such as Colombia or northern Brazil, showed an increased circulation of EBV2 where its prevalence rose to 20% of healthy carriers [25,26]. In this work, viral type distribution among the Argentinian isolates showed a higher prevalence of EBV1 (72%) over EBV2 (26%) genomes; however, the proportion of EBV2 genomes was higher than that previously reported for our country and for other regions, such as Europe and North America, where EBV2 only represents about 10–15% of the circulating EBV [7,16,17]. Although all geographic regions were represented by at least one EBV2 isolate in our dataset of 278 genomes, given the restricted geographic circulation of EBV2, genomes from Africa, Southeast Asia, and South America were the most prevalent, a fact that limited the study to these regions. Recombinant genomes were characterized; however, those from Africa and Papua New Guinea were characterized as EBV1–EBV2 recombinants while the Argentine recombinant genome presented with an EBV2-type EBNA2 gene and EBV1-type EBNA3 genes, which made it, to the best of our knowledge, the first EBV2–EBV1 recombinant to be described.

In line with previous knowledge, EBV types 1 and 2 represented the primary viral differentiation and this was mainly driven by variations in the EBNA2 gene and the EBNA3 gene family [16,17,21]; however, variable positions within EBNA3-family adjacent genes BZLF1, BZLF2, and BLLF1 also contributed to viral type differentiation. Here, we identified and quantified the type segregation potential of a total of 74 variable positions in the above-mentioned genes. A similar observation was previously reported by Correia et al. for the promoter region of the BZLF1 gene, although variation in the coding region of the gene and the statistical significance of these variants to type segregation were not assessed [15]. The Zp-V3 variant of the BZLF1 gene promoter was found with higher prevalence in EBV2 genomes. Some authors have suggested that the Zp-V3 variant was originally derived from EBV2 and its presence in some EBV1 genomes was due to ancestral intertype recombinations [15,22]. Among the EBV1 isolates, the Zp-V3 variant showed higher circulation in Southeast Asia and Oceania compared to other geographic regions. On the other hand, the Zp-V3 variant was strictly associated with EBV2 genomes in Argentina and Africa, a result previously reported for Argentinean EBV2 genomes, although with a smaller number of isolates [24,27]. Additionally, differences were also found between viral types in the coding region of the BZLF1 gene. In this regard, the EBV2 genomes exclusively harbored the BZLF1-A haplotype regardless of geographic origin. On the other hand, geographic differences in Zta haplotypes were found for EBV1 genomes: while the BZLF1-A and BZLF1-B haplotypes were more prevalent in Southeast Asia, BZLF1-C prevailed in Africa and BZLF1-B in South America. All together, these results align with previous findings that reported an association between the BZLF1-A haplotype and the Zp-V3 variant in EBV2 isolates [24,28]. Furthermore, and also in line with Alves et al., we disclosed the association of the Zp-P variant with the BZLF1-B haplotype in EBV1 genomes [28]. Although the most frequent Zta haplotype in our Argentine cases was BZLF1-B, which was consistent to that reported by Alves et al. in Brazilian cases, our observation could be biased by the fact that only four infectious mononucleosis samples were included in the present series. In this regard, a previous work from our group reported that the BZLF1-C haplotype was the most prevalent among primary infection cases in Argentina [24]. The possible impact of variants in the promoter and coding region of the BZLF1 gene in the biology of the EBV warrants further investigation. Bristol et al. reported a higher lytic phase activation in EBV2-infected B cells and that this could be due to an increased expression of the Zta protein under the Zp-V3 promoter haplotype [14]. However, an increased dimerization capacity (active form) of the Zta protein in EBV2-infected cells could not be discarded, given that most variable positions in the BZLF1-A haplotype are located in this domain.

Regarding the BZLF2 gene, five missense variants in the gp42 protein were identified as exclusively harbored by EBV2, four of which were previously characterized by Correia et al. as having type-differentiation strength [16]. Notably, this study was the first to identify Val76Ile as an EBV2-specific variable position, enhancing the ability of BZLF2 to distinguish between EBV types. Since changes in Ala38Ser, Val76Ile, and Gln92Ala were located in the gp42 interaction domain with gH, which extends from amino acid residues 36 to 96 [29], and it was recently demonstrated that the interaction between gp42 and the gH/gL heterodimer defines cellular tropism [30,31], the identified amino acid changes could affect gp42–gH interaction and, hence, modulate the efficiency of EBV2’s infection of B cells. Concerning the BLLF1 gene, ten variable amino acids with type segregation force were found in the N-terminal domain of the gp350/220 protein in EBV2, all of which could affect viral infection since this region contains the binding site for the CR2 cellular receptor but also serves as epitopes for recognition by host neutralizing antibodies [32,33,34,35]. Additionally, this finding also correlated with the results reported by Luo et al., who observed a significant association between nine of the amino acid variants described here and EBV2 [36]. Again, the variant Gln67Leu was characterized here for the first time as having type segregation capacity for EBV2 isolates, further strengthening the ability of the BLLF1 gene to distinguish between EBV types. A similar observation, although not referring to specific variable positions, was also reported by Correia et al. for the N-terminal domain of gp350/220 in EBV isolates from various geographic origins [15]. The authors identified three clades in the phylogenetic reconstruction and suggested that the variability in the N-terminal region of this glycoprotein was not solely restricted to the viral type but also related to the geographic origin of viral isolates. Our results aligned with those of Correia et al. since variants with preferential circulation in OCE were also characterized. Finally, when studying intertype variability by geography, we characterized prevalent amino acid changes in EBV2 isolates from AFR but not in SAM or SEA. Examining variable positions in the gp350/220 and gp42 glycoproteins of EBV2 could be valuable for vaccine design, particularly in regions with higher EBV2 prevalence. As a deeper layer in our analysis, provided by the contribution of new SAM isolates, further differences between EBV1 and EBV2 were noted throughout the remainder of the genome, and these variations were organized into three distinct genomic blocks apart from the previously mentioned genes.

When assessing intertype variation at the genomic level in those geographic regions where EBV2 was present, EBV2 displayed less variability than EBV1 in AFR and SEA but not in SAM isolates. On the other hand, when focusing on the geographic origins of the isolates, those from SEA presented the greatest amount of variation irrespective of the viral type, suggesting a closer relation between isolates from SAM and AFR. Moreover, the distribution of variable positions along the entire genome also differed in these isolates, where AFR and SAM also showed a common variability pattern of distribution for each viral type with a higher number of fixed positions for EBV2. Altogether, these results demonstrate a closer relation between EBV isolates from SAM and AFR and that EBV2 is less diverse than EBV1.

Although variability in EBV1 has been widely studied [15,16], EBV2 variability has been studied to a lesser extent, and in particular, isolates from South America have been misrepresented. In this study, PCA allowed us to further explore the above-mentioned variability patterns and unequivocally assess their statistical significance to viral type and geographic segregation. While it discriminated EBV1 isolates from SEA, SAS, and OCE from those from the rest of the world, it also revealed the geographic structures of EBV2 isolates, segregating SAM and AFR from SEA genomes. This latter finding for EBV2 was similar to that observed in our previous study for EBV1 [19]. Here, when discriminated by viral type, both EBV1 and EBV2 genomes from SAM were segregated together with the AFR isolates and differentiated from those from SEA. These clustering patterns further reinforced the theory that the ancestral introduction of the EBV to SAM might have been influenced by slave trading from Africa during the colonial period of South America [19,37]. Furthermore, no geographically segregating positions were observed in the EBV2 EBNA3 gene family and its adjacent regions, a similar result to that of Kaymaz et al., who compared divergence levels between viral types in each gene and found a greater divergence in the EBNA3 gene family and adjacent genes among EBV1 isolates, but not in EBV2, from Kenya [22]. Similarly, EBV genes BPLF1, EBNA1, and BcRF1 contained the highest proportions of missense variants that contributed to the segregation of AFR and SAM genomes from those from SEA. Moreover, in this study, it was observed that in the three analyzed geographic regions, EBV2 genomes had fewer variable positions compared to the EBV2 reference genome than EBV1 isolates from these regions compared to the EBV1 reference. Consistent with this result, Kaymaz et al. compared divergence levels between viral types in AFR isolates and reported lower divergence in EBV2 genomes [22].

Most of the efforts to develop prophylactic vaccines against the EBV have focused on the gp350 protein, the most abundant envelope protein encoded by the BLLF1 gene. These approaches have utilized a variety of designs, including purified or recombinant envelope proteins, virus-like particles, mRNA, and nanoparticles [31,38,39,40,41]. More recently, combining gp350 with other proteins [40], researchers found an abundant envelope protein encoded by the BLLF1 gene response. Currently, two phase 1 clinical trials have been launched to evaluate EBV vaccine candidates [42]. The first involves a ferritin-based nanoparticle vaccine displaying gp350 (NCT04645147) while the second involves an mRNA vaccine (mRNA-1189) that includes four mRNAs encoding gH, gL, gp42, and gp220 (NCT05164094). In addition, therapeutic EBV vaccines targeting virus-associated malignancies, such as nasopharyngeal carcinoma, have been also explored using various designs. Most of these efforts have focused on the LMP2 and EBNA1 proteins, although other antigens, including EBNA3C and Zta, have also been studied [31,39,41,43].

Understanding how EBV genome variation correlates with geographic location could not only be essential in clarifying the virus’s role in disease development but also support efforts toward creating an effective EBV vaccine, in particular in those regions where both EBV types are prevalent. Addressing this complexity will be critical to ensure broad and reliable protection against the EBV. Ongoing initiatives to create an EBV vaccine and/or antiviral treatments should incorporate knowledge in these variations to improve their efficacy.

## 4. Methods

### 4.1. Ethic Statement

The hospital’s ethics committee, *Comité de Ética en Investigación (CEI)* reviewed and approved this study (CEI No. 17.25) according to the human experimentation guidelines of our institution and the Helsinki Declaration of 1975 as amended in 1983. Clinical samples from patients with EBV infection were anonymized before the study. Written informed consent was obtained from all patients’ parents or legal guardians.

### 4.2. Patients and Samples

This study included samples from 45 patients with EBV-associated pathologies from Argentina: (i) 4 pediatric patients with infectious mononucleosis (IM), median age of 5.75 years (range 3 to 9 years), 75% male; (ii) 4 pediatric patients with immunodeficiency, median age of 5 years (range 3 to 12 years), 50% male (of the patients included in this group, 3 had secondary immunodeficiency following a liver transplant and one had a diagnosis of inborn error of immunity); (iii) 35 pediatric patients with EBV-positive lymphomas (22 HL, 6 non-Hodgkin lymphomas, and 7 PTLDs, median age of 9 years (range 3 to 14 years), 65.7% male; and (iv) 2 asymptomatic pregnant adult healthy carriers (24 and 29 years). The type of sample used in each case is indicated in Appendix A.

### 4.3. DNA Extraction and Viral Load Analysis

Peripheral blood mononuclear cells (PBMCs) were isolated from whole blood with Ficoll-Paque (GE Healthcare, GE17-1440-02, Chicago, IL, USA). Total DNA was purified from PBMCs, pharyngeal secretions, saliva, and fresh tumor biopsies using QIAamp DNA Mini Kit (QIAGEN, 51304, Hilden, Germany) according to the manufacturer’s instructions.

The number of EBV genome copies was measured in all samples using a quantitative real-time PCR (qPCR) assay on a LightCycler 480 device as previously described [19]. Samples with a viral load above 500,000 EBV copies/μg of total genomic DNA were selected for targeted NGS sequencing.

### 4.4. Library Preparation and NGS Sequencing on the Illumina Platform

Sequencing libraries were constructed using SureSelect QXT Target Enrichment Kit (Agilent Technologies, G9683B, Santa Clara, CA, USA) and RNA capture probes specifically designed for EBV (Agilent Technologies, 5190-4806) following the manufacturer’s instructions and as previously described [19]. Finally, sequencing of the libraries was performed on the Illumina NexSeq500 system. The *.fastq* files were deposited in the NCBI database under the BioProject accession number PRJNA679281 (see Appendix A).

### 4.5. Bioinformatics, Viral Typing and Recombination Analysis Between Viral Types

To study the variability of isolates from Argentina within the geographical context, 242 *.fastq* files corresponding to EBV genomes isolated from various geographical regions (79 from Africa, 19 from Australia, 33 from Europe, 19 from North America, 3 from South America, 34 from Oceania, 5 from South Asia, and 50 from Southeast Asia) were downloaded from the SRA-NCBI database, corresponding to the Bioproject numbers PRJNA522388, PRJNA505149, PRJNA552587, PRJNA480052, and PRJEB2768 (see Appendix A). The 287 *.fastq* files (45 generated and 242 downloaded) were analyzed using a bioinformatic pipeline based on a mapping strategy as previously described [19]. In this way, 9 genomes (6 generated and 3 downloaded) with a coverage of less than 50% were excluded from all further analysis. Since linkage disequilibrium was described between variants in EBNA2 gene and EBNA3 family of genes [16,44], and since these variants define EBV type, the remaining 278 genomes were typed based on coverage and the number of variants in these genes after mapping the reads against both viral reference genomes (NC_007605.1 for EBV1 and NC_009334.1 for EBV2) [19]. For those viral genomes showing evidence of potential recombination events between viral types, de novo assembly was performed using *Spades v.3.14.0* [45]. Scaffolds were formed from the assembly of reads, which were aligned and ordered using the *ABACAS program v.1.3.1* [46]. Consensus sequences constructed in this way were aligned with the reference genomes of both viral types using *MAFFT v.7.450* [47]. Phylogenetic reconstruction was performed from the coding region of the EBNA2 and EBNA3A genes using *IQ-TREE v.1.6* [48]. The statistical validation of each tree’s topology was conducted with 1000 replicates using the ultrafast bootstrap method. The phylogenetic trees were visualized using the online software iTOL *v.7* (https://itol.embl.de/, accessed on 12 June 2023). Additionally, recombination signals were studied using nine different computational algorithms with *RDP4 program v.4.97* and the detection of a recombination event was considered significant when detected by at least three independent computational algorithms [49].

### 4.6. Bioinformatic Pipeline for the Construction of Multi-Sample .vcf Files

The quality of the reads was assessed using *FastQC v.0.12.1* (http://www.bioinformatics.babraham.ac.uk/projects/fastqc accessed on 12 June 2023). The programs *fastp v.0.20.1* [50] and *PRINSEQ v.0.20.4* [51] were employed to remove low-quality ends, low-quality reads, PCR duplicates, and adapters. The reads were aligned to the reference genome of EBV1 using *BWA v.0.7.17-r1188x* [52]. *SAMTools v1.9* [53] was then utilized to convert the resulting *.sam* files to *.bam* files. The intersect-v function of *BedTools v2.29.2* [54] was used to remove reads aligning to repetitive regions and OriLyt. From the obtained *.bam* files, *.gvcf* files were generated using the *HaplotypeCaller* tool of *GATK v4.1* (Genome Analysis Toolkit) [55]. Joint genotyping was performed from the *.gvcf* files of each sample using *GATK v4.1 GenotypeGVCFs*. The next step involved extracting variable positions, including both SNPs and INDELs, using the *SelectVariants* tool of *GATK v4.1*, to obtain a *multi-sample.vcf* file. Subsequently, *VariantFiltration* and the *bcftools* filter function of *GATK v4.1* were employed to identify and extract viral variants that did not meet the established quality criteria; (i) variable positions, present in fewer than five genomes, and (ii) variable positions covered in less than 50% of the total genome (missing data) were removed using the *bcftools* filter tool. The aforementioned bioinformatic strategy was employed to construct a *multi-sample_all genomes .vcf* file based on the EBV1 reference genome consisting of 278 viral genomes. From this *.vcf* file, a second *multi-sample_EBV1.vcf* file exclusively composed of pre-typed EBV1 genomes was constructed to study the segregation and geographic variability of this viral type. On the other hand, to study the variability of EBV2, a new *multi-sample_EBV2.vcf* file was constructed using the reference genome for this viral type and the genomes previously typed as EBV2. Finally, to study intertype variability in each geographical region separately, data from the original *multi-sample_all genomes.vcf* file, constructed from the NC_007605.1 reference and 278 genomes were used to construct three region-specific *multi-sample.vcf* files (Africa, South America, and Southeast Asia). In all cases, the *SnpEff v5.0* program was utilized for variant annotation and prediction of their effect on viral proteins [56]. In summary, the bioinformatic pipeline used in this study and the files obtained in each step are described in Figure 6.

### 4.7. Principal Component Analysis

Using the *multi-sample.vcf* files as input, the *adegenet package v2.1.10* [57] was applied for Principal Component Analysis (PCA). The *vcfR package v1.14.0* [58] was employed for *.vcf* file manipulation.

### 4.8. Variant Frequency

From the annotated *multi-sample.vcf* files, the frequency of variation at each position of the EBV genome relative to the reference was assessed using *VCFTools v0.1.15* [59]. Comparisons were made between viral types and geographies to identify positions contributing to differentiation. When referring to variable positions in the BZLF1 gene or its protein product Zta, BZLF1 haplotypes were termed using the classification introduced by Luo et al. [60], and when referring to variable positions within the BZLF1 promoter region, haplotypes were termed according to the classification introduced by Gutierrez et al. [61].

### 4.9. Statistical Analysis

Statistical analysis was performed using R software, version 4.0.2 [62] Normality was initially evaluated using the Shapiro test to compare distributions between groups. Normality assumption was rejected if at least one group had a *p* value < 0.05, in which case the Kruskal–Wallis test was applied for comparisons among more than two groups and the Wilcoxon test was used for pairwise comparisons. Fisher’s exact test was used to assess association between categorical variables. All *p* values from multiple comparisons were adjusted using the Bonferroni correction. A *p* value ≤ 0.05 was considered statistically significant.

### 4.10. Data Availability

The NGS data generated in this project are available at the Sequence Read Archive (SRA) at the NCBI under Bioproject No. PRJNA679281. All other Appendix A are available at https://ri.conicet.gov.ar/handle/11336/248073 accessed on 27 January 2025.

## Figures and Tables

**Figure 1 ijms-26-02708-f001:**
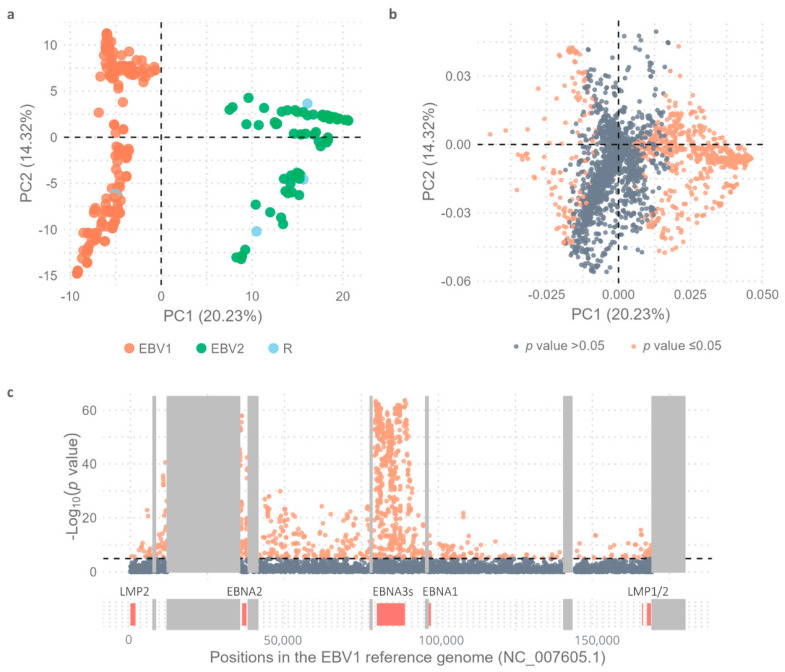
Principal Component Analysis. (**a**) Distribution of viral genomes based on PC1 (20.23%) and PC2 (14.32%); EBV1 (orange), EBV2 (green), and viral-type recombinants (R) (in light blue). (**b**) Dots represent the 4.541 variable positions; orange dots represent statistically significant values and gray dots non-statistically significant values as obtained by comparing the allelic frequencies of EBV1 and EBV2 using Fisher’s exact test and subsequent Bonferroni correction. (**c**) Distribution of identified variants along the genome based on the −log_10_(*p* value) when comparing the allelic frequencies of EBV1 and EBV2 using Fisher’s exact test and subsequent Bonferroni correction; orange dots represent statistically significant values and gray dots non-statistically significant values. Black dotted line represents the threshold value above which the *p* values are considered significant [−log_10_(*p* value) = 4.95]. Most relevant genes are represented below, according to their positions in the genome. Gray boxes represent the repetitive regions of the viral genome.

**Figure 2 ijms-26-02708-f002:**
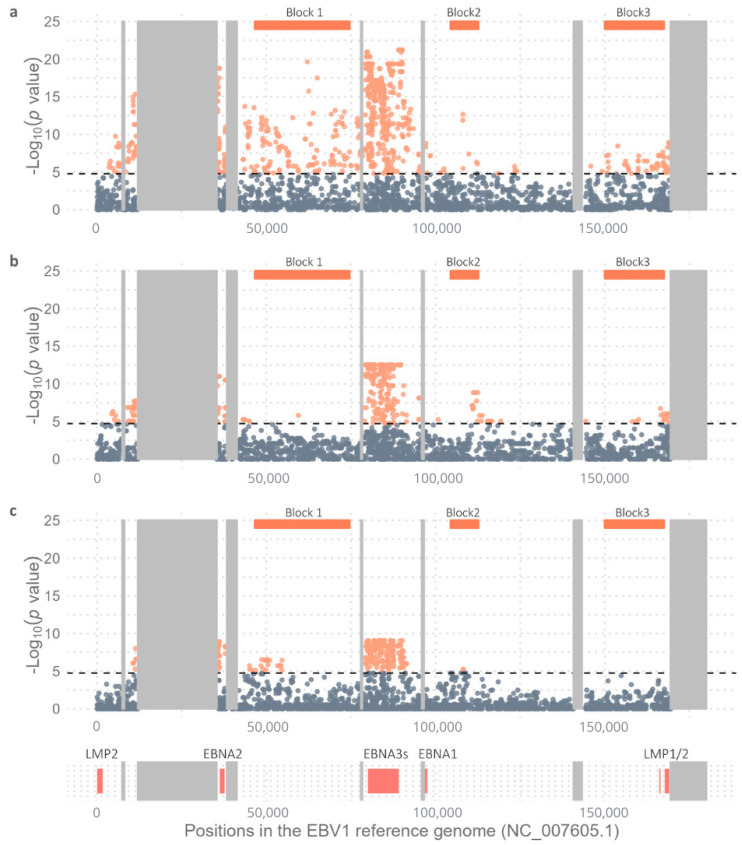
Viral type variability in (**a**) Africa, (**b**) Southeast Asia and (**c**) South America. Figure shows distribution of identified variants along the genome based on the −log_10_(*p* value) when comparing the allelic frequencies of EBV1 and EBV2 using Fisher’s exact test and subsequent Bonferroni correction; orange dots represent statistically significant values and gray dots non-statistically significant values. Black dotted line represents the threshold value above which the *p* values are considered significant [−log_10_(*p* value) AFR = 4.78; −log_10_(*p* value) SEA = 4.73 and −log_10_(*p* value) SAM = 4.75]. Most relevant genes are represented below according to their positions in the genome. Orange boxes above the graph delimit the 3 most informative blocks in terms of variable positions. Most relevant viral genes are represented as coral boxes below the graph according to their positions in the genome. Gray boxes represent the repetitive regions of the viral genome.

**Figure 3 ijms-26-02708-f003:**
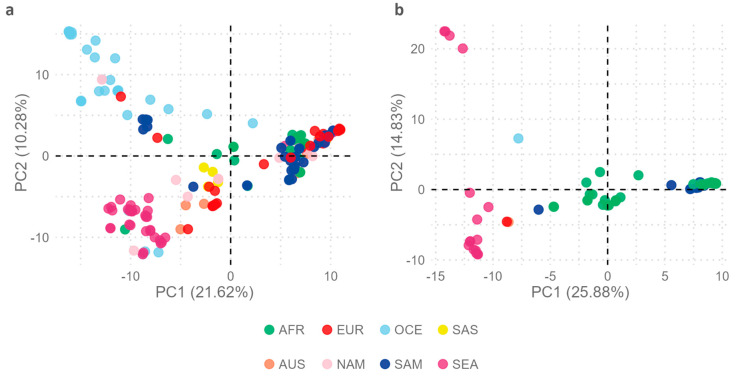
Principal Component Analysis. Genomes have been colored according to the geographic origins of the isolates. (**a**) Distribution of EBV1 genomes in relation to PC1 (21.62%) and PC2 (10.28%). (**b**) Distribution of EBV2 genomes in relation to PC1 (25.88%) and PC2 (14.83%).

**Figure 4 ijms-26-02708-f004:**
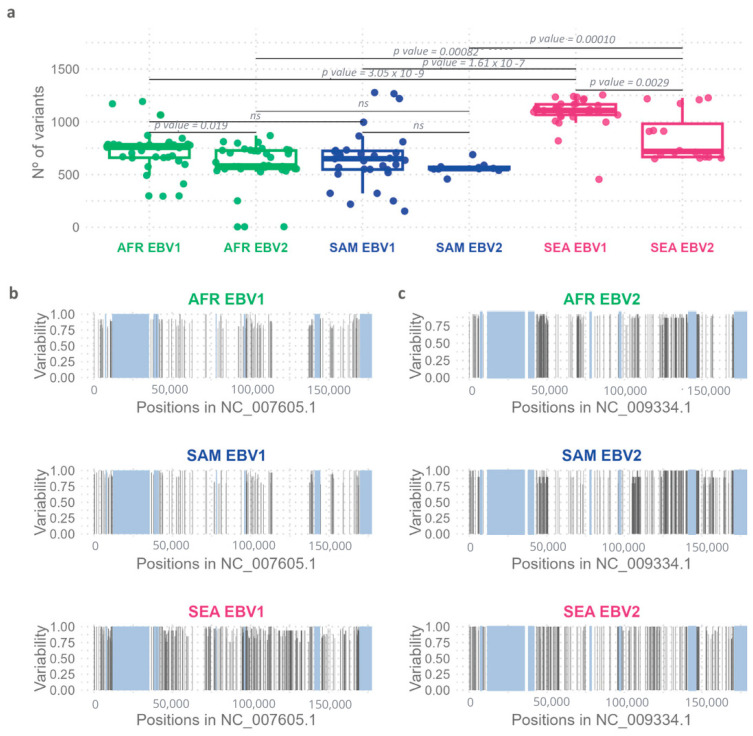
Genomic variability of EBV1 and EBV2 in relation to geographic origin. (**a**) Quantification of the number of variants in isolates from different geographic regions relative to each reference genome. The boxes represent the interquartile range, with the inner line indicating the median value for each group and the whiskers showing the maximum and minimum values. (**b**) Distribution of common variants by geography along the EBV1 reference genome (NC_007605.1). Gray boxes represent the repetitive regions. (**c**) Distribution of common variants by geography along the EBV2 reference genome (NC_009334.1). Gray boxes represent the repetitive regions.

**Figure 5 ijms-26-02708-f005:**
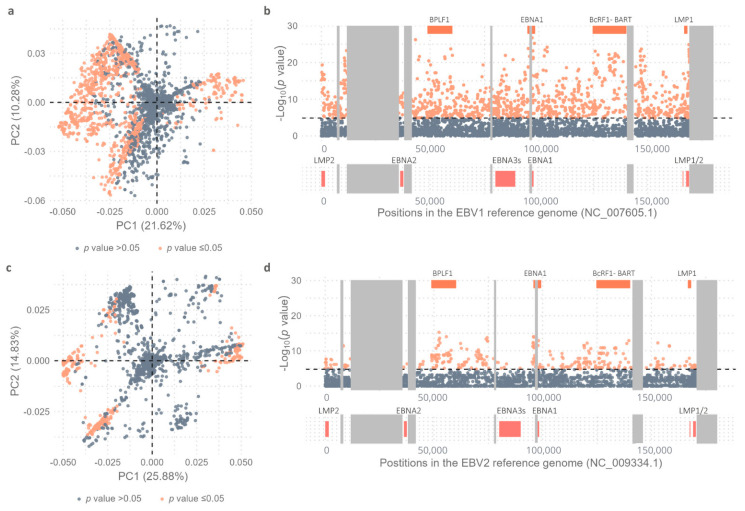
Variability Study. (**a**) Dots represent the 3666 variable positions; orange dots represent statistically significant values and gray dots non-statistically significant values as obtained by comparing the allelic frequencies of SEA, SAS, and OCE with EUR, AFR, AUS, NAM, and SAM using Fisher’s exact test and subsequent Bonferroni correction. (**b**) Distribution of identified variants along the genome based on the −log_10_(*p* value) when comparing the allelic frequencies of SEA, SAS, and OCE with EUR, AFR, AUS, NAM, and SAM using Fisher’s exact test and subsequent Bonferroni correction; orange dots represent statistically significant values and gray dots non-statistically significant values. Black dotted line represents the threshold value above which the *p* values are considered significant [−log_10_(*p* value) = 4.86]. Orange boxes above the graph represent the 4 most informative coding regions in terms of variable positions. Most relevant viral genes are represented as coral boxes below the graph according to their positions in the genome. Gray boxes represent the repetitive regions of the viral genome. (**c**) Dots represent the 3276 variable positions; orange dots represent statistically significant values and gray dots non-statistically significant values as obtained by comparing the allelic frequencies of SEA and OCE with AFR and SAM using Fisher’s exact test and subsequent Bonferroni correction. (**d**) Distribution of identified variants along the genome based on the −log_10_(*p* value) when comparing the allelic frequencies of SEA and OCE with AFR and SAM using Fisher’s exact test and subsequent Bonferroni correction; orange dots represent statistically significant values and gray dots non-statistically significant values. Black dotted line represents the threshold value above which the *p* values are considered significant [−log_10_(*p* value) = 4.74]. Orange boxes above the graph represent the 4 most informative coding regions in terms of variable positions. Most relevant viral genes are represented as coral boxes below the graph according to their positions in the genome. Gray boxes represent the repetitive regions of the viral genome.

**Figure 6 ijms-26-02708-f006:**
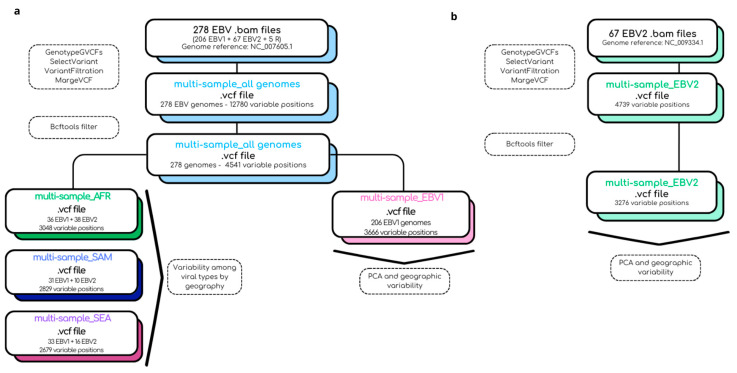
Summary of the bioinformatic pipeline used for the construction of *multi-sample.vcf* files. (**a**) The reads were aligned to the EBV1 reference genome (NC_007605.1) to generate a multi-sample .vcf file consisting of 278 viral genomes. From this file, the corresponding subsets for EBV1 and geographic regions were then generated. (**b**) The reads were aligned to the EBV2 reference genome (NC_009334.1) to generate a multi-sample .vcf file consisting of 67 EBV2 viral genomes.

## Data Availability

Data are contained within the article.

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
