# Peer review of "A Comparative Genomic Analysis of Epstein–Barr Virus Strains with a Focus on EBV2 Variability"

_ijms, 2025, doi:10.3390/ijms26062708_

Round 1

Reviewer 1 Report

Comments and Suggestions for Authors

The manuscript “Comparative Genomic Analysis of Epstein-Barr Virus Strains 2 with a focus on EBV2 variability” by Ana Catalina Blazquez et al. describes an in-depth statistical study of the differences of between Epstein-Barr viruses EBV1 and EBV2 with a focus on EBV2. The reason for this focus is that EBV1 has been studied considerably more while with EBV2 it has been difficult to obtain sequence data due to restricted geographic access. The rational for this study is not only to understand the differences but to obtain data to better treat individuals with the different strains. The authors provide significant data comparing the different regions of the genomes regarding mutations found in these regions including separating this data into various geographic locations. There are also some discussions on how these mutations can potentially affect epidemiological outcomes in the various geographic regions. The paper is well written and contains considerable data that can be applied to future studies.

Minor issue: Figure 1 should probably be moved to be after its first reference in the rather than near the end. Also, Figure 3 should reference its parts a, b, and c in the text rather than just referring to this figure as Figure 3.

Author Response

Reviewer 1 Comments
Comments to the Authors
The manuscript “Comparative Genomic Analysis of Epstein-Barr Virus Strains 2 with a focus on EBV2 variability” by Ana Catalina Blazquez et al. describes an in-depth statistical study of the differences of between Epstein-Barr viruses EBV1 and EBV2 with a focus on EBV2. The reason for this focus is that EBV1 has been studied considerably more while with EBV2 it has been difficult to obtain sequence data due to restricted geographic access. The rational for this study is not only to understand the differences but to obtain data to better treat individuals with the different strains. The authors provide significant data comparing the different regions of the genomes regarding mutations found in these regions including separating this data into various geographic locations. There are also some discussions on how these mutations can potentially affect epidemiological outcomes in the various geographic regions. The paper is well written and contains considerable data that can be applied to future studies.

Minor issue: Figure 1 should probably be moved to be after its first reference in the rather than near the end. Also, Figure 3 should reference its parts a, b, and c in the text rather than just referring to this figure as Figure 3.

Response: We noticed that the manuscript was reorganized to comply with the journal's guidelines, which resulted in the Methods section being moved to the end. As a result, the original numbering of the figures and supplementary tables has been updated to reflect this new order. We have also reflected on this change in the repository.

Regarding 'Figure 1,' it originally corresponded to the Methods section but is now labeled as 'Figure 6.' We have also expanded the figure legend to provide a clearer description of parts A and B. Since we reference this figure in the Results section (line 118), we have updated the text to read: '(See Figure 6 in Methods).

Regarding the reviewer's suggestion about the former Figure 3 (now Figure 2), we believe that the purpose of this figure is to compare the variability blocks/islands across the three geographic regions: a) Africa, b) Southeast Asia, and c) South America. For this reason, it is not feasible to address these regions separately, as the figure is intended to be analyzed as a whole.

Reviewer 2 Report

Comments and Suggestions for Authors

The authors used the Next Generation Sequencing approach to decipher Epstein-Bar Virus Strains (EBV) using comparative genomic analysis with special emphasis on accounting variability in the EBV2 genome owing to its restricted geographic presence. This is an especially intriguing premise, given the potential economic impact of EBV-associated diseases in developing more efficient vaccines in the future. The well-organized text offers a solid review that lays the foundation for vaccine designs owing to the genomic variations based on the geography and their possible significant implications. However, there are a few essential points that warrant further attention.

1)      Line {14-16} In this study, the authors sequenced and analyzed 28 EBV1 and 10 EBV2 genomes from Argentina, combined with 239 publicly available complete genomes from other geographic regions. This resulted in an initial multi-sample VCF file comprising 278 EBV genomes. However, the total number adds up to 277 rather than 278. Please clarify this discrepancy.

2)      Please re-structure legends for Figure 4 and Figure 5.

3)      The discussion should also focus on genomic variability and their prospective implications on vaccine development.

Author Response

Reviewer 2 Comments
Comments and Suggestions for Authors

The authors used the Next Generation Sequencing approach to decipher Epstein-Bar Virus Strains (EBV) using comparative genomic analysis with special emphasis on accounting variability in the EBV2 genome owing to its restricted geographic presence. This is an especially intriguing premise, given the potential economic impact of EBV-associated diseases in developing more efficient vaccines in the future. The well-organized text offers a solid review that lays the foundation for vaccine designs owing to the genomic variations based on the geography and their possible significant implications. However, there are a few essential points that warrant further attention.

1) Line {14-16} In this study, the authors sequenced and analyzed 28 EBV1 and 10 EBV2 genomes from Argentina, combined with 239 publicly available complete genomes from other geographic regions. This resulted in an initial multi-sample VCF file comprising 278 EBV genomes. However, the total number adds up to 277 rather than 278. Please clarify this discrepancy.

Response: We acknowledge the reviewer's observation regarding the unclear explanation of the number of genomes included in the study, particularly the omission of the potential recombinant genome from Argentina in the abstract. To address this, we have revised line 15 of the abstract to include this information. The updated paragraph now reads: “In this study, we sequenced and analyzed 28 EBV1 genomes, 10 EBV2 genomes, and a potential recombinant genome from Argentina. These were combined with 239 complete genomes downloaded from other geographic regions, resulting in an initial multi-sample VCF file comprising 278 EBV genomes.”

2) Please re-structure legends for Figure 4 and Figure 5.

Response: We appreciate the reviewer's recommendation and have restructured the legends for Figures 4 and 5.

Regarding the former Figure 4 (now Figure 3), the updated legend (lines 287–290) in the revised version reads as follows: “Principal Component Analysis: Genomes were color-coded based on the geographic origin of the isolates. (a) Distribution of EBV1 genomes in relation to PC1 (21.62%) and PC2 (10.28%). (b) Distribution of EBV2 genomes in relation to PC1 (25.88%) and PC2 (14.83%).”

Regarding the former Figure 5 (now Figure 4), the updated legend (lines 320–328) in the revised version reads as follows: “Genomic variability of EBV1 and EBV2 in relation to geographic origin. (a) Quantification of the number of variants in isolates from different geographic regions relative to each reference genome. The boxes represent the interquartile range, with the inner line indicating the median value for each group, and the whiskers showing the maximum and minimum values. (b) Distribution of common variants by geography along the EBV1 reference genome (NC_007605.1). Gray boxes represent the repetitive regions. (c) Distribution of common variants by geography along the EBV2 reference genome (NC_009334.1). Gray boxes represent the repetitive regions.”

3) The discussion should also focus on genomic variability and their prospective implications on vaccine development.

Response: We agree with the reviewer that the prospective implications of EBV genome variability on vaccine development were not adequately addressed in the Discussion section. To address this, we have included a new paragraph at the end of the Discussion section (lines 515–530 and 534–535).

The revised manuscript now states: “Most of the efforts to develop prophylactic vaccines against EBV have focused on the gp350 protein, the most abundant envelope protein encoded by the BLLF1 gene. These approaches have utilized a variety of designs, including purified or recombinant envelope proteins, virus-like particles, mRNA, and nanoparticles (Jean-Pierre et al., 2021; Rühl et al., 2020; Sun et al., 2021; Tornesello et al., 2022; Zhong et al., 2022). More recently, combining gp350 with other proteins—primarily gH, gL, and gp42, encoded by the BZLF2 gene—has shown promise in achieving a more robust protective response. Currently, two phase 1 clinical trials have been launched to evaluate EBV vaccine candidates (Zhong et al., 2024). The first involves a ferritin-based nanoparticle vaccine displaying gp350 (NCT04645147), while the second is an mRNA vaccine (mRNA-1189) that includes four mRNAs encoding gH, gL, gp42, and gp220 (NCT05164094). In addition, therapeutic EBV vaccines targeting virus-associated malignancies, such as nasopharyngeal carcinoma, have been also explored using various designs. Most of these efforts focus on the LMP2 and EBNA1 proteins, although other antigens, including EBNA3C and Zta, have also been studied  (Cohen, 2024; Jean-Pierre et al., 2021; Rühl et al., 2020; Tornesello et al., 2022).”

We have also included the concluding statement: “Addressing this complexity will be critical to ensuring broad and reliable protection against EBV.”

As a result of these additions, we have also incorporated new references.